# Yield and Nutraceutical Value of Lettuce and Basil Improved by a Microbial Inoculum in Greenhouse Experiments

**DOI:** 10.3390/plants12081700

**Published:** 2023-04-19

**Authors:** Elia Pagliarini, Francesca Gaggìa, Maurizio Quartieri, Moreno Toselli, Diana Di Gioia

**Affiliations:** Department of Agricultural and Food Science, University of Bologna, V. le Fanin 44, 40127 Bologna, Italy

**Keywords:** PGPR, *Bacillus* spp., *Lactuca sativa* L., *Ocinum basilicum* L., mineral uptake, greenhouse

## Abstract

Members of *Bacillus* spp. have been widely used to enrich the soil/root interface to provide plant growth promoting activities. A new isolate, namely to *Bacillus* sp. VWC18, has been tested under greenhouse conditions in lettuce (*Lactuca sativa* L.) pots at different concentrations (10^3^, 10^5^, 10^7^, and 10^9^ CFU·mL^−1^) and application time (single inoculum at transplant and multiple inoculum every ten days) to evaluate the best application dose and frequency. Analysis of foliar yield, main nutrients, and minerals evidenced a significant response for all applications. The lowest (10^3^ CFU·mL^−1^) and the highest doses (10^9^ CFU·mL^−1^), applied every ten days until harvest, had the greatest efficacy; the nutrient yield (N, K, P, Na, Ca, Fe, Mg, Mn, Cu, and B) increased more than twice. A new randomized block design with three replicates was then performed in lettuce and basil (*Ocinum basilicum* L.), with the two best performing concentrations applied every ten days. In addition to previous analysis, root weight, chlorophyll, and carotenoids were also examined. Both experiments confirmed the previous results: inoculation of the substrate with *Bacillus* sp. VWC18 promoted plant growth, chlorophyll, and mineral uptake in both crop species. Root weight duplicated or triplicated compared to control plants, and chlorophyll concentration reached even higher values. Both parameters had a dose-dependent increase.

## 1. Introduction

The rhizosphere represents a unique ecological niche for several microorganisms known as Plant Growth-Promoting Rhizobacteria (PGPR) [1]. Many bacterial strains (i.e., *Bacillus* spp., *Pseudomonas* spp.) play important functions in plant nutrition [2,3]; they can provide a wide range of activities, improve crop yield and productivity [4], and counteract phytopathogens [5]. Thanks to their metabolic and catabolic functions, they act as mineral solubilizer, nitrogen fixing, iron capture, phytohormone, and antibiotic producers [6,7]. However, massive agricultural production and unsustainable land use practices that overuse synthetic fertilizers and pesticides have impoverished microbial life in soil [8,9] and, as a consequence, its ability to interact with plant roots and physiology. *Bacillus* spp. comprise several strains with PGP attributes for rhizosphere enrichment/manipulation [10,11]; inoculation with such microorganisms can reduce or remove the requirements for chemical fertilizers or pesticides [12]. Moreover, they are spore formers, and this property is helpful for biotic stress resistance and the development of a commercial product [13]. Numerous studies have been performed with several *Bacillus* strains. Focusing on PGPR inoculation on lettuce and basil, the main achievements of recent studies include increased productivity and mineral uptake, as well as improved resistance to salt stresses and root development [14,15,16]. Interestingly, the use of PGPR inocula in different basil cultivars led to an increase in most of the growth parameters, such as fresh/dry shoot yield, dry leaf yield, and leaf area index, as well as an improved essential oil yield [17,18,19].

Lettuce is traditionally used as a model crop because of its short growth cycle; it is a very popular vegetable with a worldwide production of more than 29 million tons each year [20]. From a nutritional point of view, lettuce has an important role, as it contains vitamins and several micronutrients [21]. Basil is among the most popular herbs grown in gardens and in aromatic plant-producing companies throughout the world [16]. It is consumed fresh, dried, and processed for flavoring and for medical purposes [22]. It is the most popular fresh herb in Europe and covers between 60 and 75 percent of the total consumption of culinary herbs [23].

Growing environmental and ecological awareness has led to an increasing number of farmers adopting new sustainable and efficient cultivation systems. The human diet is more and more dependent on processed foods and less dependent on plant-based foods, which are the source of important minerals and bioactive compounds that are necessary to maintain health and prevent diseases [24]. Greenhouse cultivation needs to produce high quality and vigorous vegetable/flower seedlings, both for the market and for field planting. Green technologies, conveniently investigated, can offer an integrated approach to stepping away from the use of fertilizers and pesticides while increasing some of the mineral elements often lacking or not adequately present in human diets (i.e., calcium (Ca), magnesium (Mg), iodine (I), zinc (Zn), selenium (Se), iron (Fe), copper (Cu), and silicon (Si)) [24].

The present study was designed to evaluate the impact of a *Bacillus* strain (*Bacillus* sp. VWC18), recently isolated from vegetable waste compost, on edible crops, namely lettuce and basil. We hypothesized that *Bacillus* sp. VWC18 acts as a PGPR, thus improving yield and increasing the quality parameters essential for human health. The inoculum concentration in most applications is frequently fixed in the range of 10^6^ CFU·mL^−1^–10^9^ CFU·mL^−1^ [5,14,15], and one application is usually performed before harvest [25,26,27]; only few studies have investigated the efficacy of different concentrations or even different application times [5,28,29], largely because it is a time-consuming and cumbersome process. Our study is focused on the use of different inoculum concentrations and application frequencies in order to select the best rhizosphere treatment for commercial and field plant production. 

## 2. Results

### 2.1. Preliminary Greenhouse Experiment

#### 2.1.1. Growth and Yield Parameters of Lettuce

The effects of the single (S) and multiple (M) microbial inocula used in the preliminary greenhouse experiment on lettuce plants were evaluated by measuring the following parameters: fresh and dry leaf biomass (FW and DW) and the percentage of total dry matter (DM%) (Figure 1a–f). All plants receiving the single microbial inoculum at different concentrations evidenced higher FW values compared to S-CTR (40.83 ± 4.36 g·plant^−1^) (Figure 1a); in particular, S-B1 and S-B2 registered significantly higher values, 51.60 ± 8.63 g·plant^−1^ and 50.39 ± 6.50 g·plant^−1^, respectively. The DW was significantly increased in S-B2 (Figure 1b). S-CTR showed the greatest DM%, but no significant differences were reported among values (Figure 1c). With respect to the multiple inocula, plants receiving the microbial enrichment evidenced a significant difference in FW, DW, and DM% measurements compared to M-CTR (Figure 1d–f). The FW of M-B1 and M-B4 increased significantly (more than 40%), with a value of 65.70 ± 7.50 g·plant^−1^ and 67.20 ± 8.27 g·plant^−1^, respectively. Likewise, the DW was significantly different (*p* ≤ 0.05), with higher values compared to M-CTR; DM% was higher and more significant in M-CTR compared to M-B4 (7.57 ± 0.37% vs. 6.67 ± 0.74%; *p* ≤ 0.05). M-B1, M-B2, and M-B3 registered intermediate values for DM% and belonged to the same significance groups (7.34 ± 1.00%, 7.34 ± 0.51%, 7.21 ± 0.76%).

#### 2.1.2. Mineral Uptake: N, P K, Na, Ca, Mg, Mn, Fe, Zn, Cu, B

The effect of the two experiments (single and multiple inocula) on N, P, and K content is shown in Table 1.

The single treatment did not show significant differences in N, P, and K content among S-CTR, S-B1, S-B2, S-B3, and S-B4, although all mean values were higher in the treated plants. The multiple inoculum showed significant differences (*p* ≤ 0.01 and *p* ≤ 0.001); some essential nutrients were doubled and almost tripled compared to M-CTR (116.97 ± 38.79 for N, 12.99 ± 3.62 for P, and 57.15 ± 13.87 mg·plant^−1^ for K). In particular, M-B1 and M-B4 registered higher values (165.66 ± 19.87 and 170.49 ± 18.39 mg·plant^−1^ for N; 34.49 ± 4.66 and 26.81 ± 2.17 mg·plant^−1^ for P; 136.41 ± 11.50 and 129.05 ± 19.26 mg·plant^−1^ for K). Comparing S-B1, S-B2, S-B3, and S-B4 to M-B1, M-B2, M-B3, and M-B4, the increase was significant for N, P, and K, with a twofold increase for P. The additional minerals are shown in Table 2. Except for Na in S-B1, lettuce receiving the single inoculum did not show any significant difference compared to S-CTR, although all treatments registered the highest values. In the multiple inoculum, the obtained measures evidenced a significant increase (*p* ≤ 0.001) of over 50% in all treated plants. The treatments M-B1 and M-B4 registered the best values for all minerals, which were almost double those of M-CTR. Between the single and multiple inocula, S-CTR and M-CTR had comparable values, whereas a general increase was observed for M-B1, M-B2, M-B3, and M-B4 compared to S-B1, S-B2, S-B3, and S-B4.

### 2.2. Second Greenhouse Experiment: Lettuce

#### 2.2.1. Growth and Yield Parameters

The experiment was performed using the conditions previously optimized: the multiple inoculum at the two best performing doses. Figure 2 shows the FW, DW, and DM% of lettuce plants (leaves and roots) at the end of the growth cycle (31 days).

The two tested microbial doses, L-B1 (10^3^ CFU·mL^−1^) and L-B2 (10^9^ CFU·mL^−1^), brought a significant increase (*p* ≤ 0.001) in leaf and root FW and DW compared to L-CTR. Leaf FW increased by 32% and 28% and root FW increased by between 86% and 200%. The DM% did not show any change in the aboveground biomass, whereas root DM% was significantly different among treatments, with the highest value in L-CTR (25.54 ± 7.22%; *p* ≤ 0.05). Table 3 shows the chlorophyll concentration (a and b) and carotenoid content obtained from leaf extracts. Although there was an evident increase, the values related to chlorophyll-*a*, chlorophyll-*b*, and carotenoids in L-B1 (103.18 ± 13.68 µg·g^−1^, 46.15 ± 5.50 µg·g^−1^, 7147.74 ± 533.33 µg·g^−1^) were not significant compared to L-CTR (42.29 ± 4.42 µg·g^−1^, 18.54 ± 9.20 µg·g^−1^, 4310.03 ± 101.84 µg·g^−1^). Lettuce plants treated with the higher *Bacillus* dose (L-B2) evidenced significantly higher values (*p* ≤ 0.05) (335.76 ± 135.55 µg·g^−1^, 172.23 ± 37.93 µg·g^−1^, and 21174.75 ± 9445.08 µg·g^−1^) compared to L-CTR.

#### 2.2.2. Mineral Uptake: N, P K, Na, Ca, Mg, Mn, Fe, Zn, Cu, B, Cr, Mo, Se

In the second greenhouse, an experiment with a wider array of minerals was performed (Table 4). In particular, Cr, Mo, and Se were also taken into consideration.

Except for Mo in L-B1 plants, all values registered in lettuce subjected to the microbial treatments were significantly higher (*p* ≤ 0.001, 0.01 and 0.05) compared to the control; no significant difference was observed between the two doses of VWC18, except for Mg and Mn, which showed significantly higher values in L-B2 than L-B1. Compared to the L-CTR, N, P, and K had an increase of up to 40% (N), 100% (P), and 132% (K). The remaining elements had an increase higher than 50%, with some reaching (e.g., Na, Mg, Fe, Zn, B, Mo) or exceeding (e.g., Mn) an almost 100% increase.

### 2.3. Second Greenhouse Experiment: Basil

#### 2.3.1. Growth and Yield Parameters

Figure 3 shows the mean values related to the FW, DW, and DM% of basil plants (leaves and roots) at the end of the growth cycle (54 days).

Increases in FW and DW were observed in treated plants, both in leaves and roots. B-B1 and B-B2 values were significantly different (*p* ≤ 0.001) versus B-CTR and between the two doses, except for root FW. Leaf FW in B-B1 was twofold higher (B-CTR, 28.52 ± 11.10 g·plant^−1^; B-B1, 69.24 ± 16.98 g·plant^−1^), and B-B2 had an even greater increase (B-CTR, 28.52 ± 11.10 g·plant^−1^; B-B2, 93.43 ± 14.71 g·plant^−1^). Similarly, root FW significantly increased from 1.29 ± 0.36 g·plant^−1^ (B-CTR) to 7.90 ± 6.78 g·plant^−1^ and 28.01 ± 13.90 g·plant^−1^ (B1 and B2, respectively). With respect to DM%, a significant increase was evidenced in leaves for B-B1 and B-B2, whereas root DM% progressively decreased in both treatments compared to B-CTR. Table 5 shows the chlorophyll and carotenoid measures related to control plants (B-CTR) and the two tested Bacillus doses. All analyzed parameters were significantly higher compared to B-CTR, with a threefold increase in B-B1 (chlorophyll-a, 120.08 ± 2.52 µg·g^−1^; chlorophyll-b, 100.62 ± 33.15 µg·g^−1^), fourfold increase in B-B2 (chlorophyll-a, 122.89 ± 4.04 µg·g^−1^; chlorophyll-b, 127.38 ± 7.20 µg·g^−1^), and fivefold increase in carotenoids (B-B1, 13.183 ± 896 µg·g^−1^; B-B2, 15.292 ± 628 µg·g^−1^).

#### 2.3.2. Mineral Uptake: N, P K, Na, Ca, Mg, Mn, Fe, Zn, Cu, B, Cr, Mo, Se

The mineral uptake of basil plants is shown in Table 6.

As for lettuce, beside N, P, and K, the following minerals were analyzed: Na, Ca, Mg, Mn, Fe, Zn, Cu, B, Cr, Mo, and Se. As shown in Table 6, all minerals in B-B1 and B-B2 registered huge and significant increase compared to B-CTR. N, P, and K increased by 485%, 99.5%, and 389% in B-B1 and 832%, 110%, and 500% in B-B2. All examined minerals were significantly higher in B-B1 and B-B2 compared to B-CTR (except for Cr in B-B1), reaching values more than tenfold higher. Comparison between the two doses of VWC18 shows statistically higher values in B-B2 than in B-B1 for all nutrients, except P, Cu, and Cr.

## 3. Discussion

The biological activity of microorganisms in soil is considered one of the main drivers of soil integrity, nutrient cycling, and plant growth [30,31]. It is widely known that several *Bacillus* species can offer benefits to soil fertility, plant growth promotion, and protection from pathogens [32,33,34,35]. Based on the 16S rRNA gene sequences, the strains used in this work had 99% similarity to the genus *Bacillus* (Accession Number: OQ053249). Further investigation will be performed to obtain identification at the species level and biochemical features. Vegetable waste compost, the isolation source of *Bacillus* sp. VWC18, represents a low-input fertilization strategy and a “reservoir” of beneficial microorganisms capable of recycling nutrients and improving plant growth through broad-spectrum actions [4].

Three main issues need to be considered when attempting an enrichment of the rhizosphere: the choice of microorganisms, the doses, and the application frequency [29]. PGPR may significantly vary in terms of their features, since their actions are often strain-specific [4]. The strain VWC18 was basically selected based on its isolation source and the fact it is a member of the *Bacillus* genus. The preliminary test carried out in this work was an experimental proof of its PGPR capabilities. In terms of mode of application, most investigations take into account either seed coating, foliar spray, soil drenching, or fertirrigation after seedling emergence at the rhizosphere level [28,29]. In this work, an inoculum at the rhizosphere level was chosen in order to better simulate the ordinary work within a horticultural nursery and field, where transplantation of plantlets is the starting point for the production of strong and vigorous compact plants for direct sale. This strategy is more convenient for operators, who only need to add the selected formula via the irrigation plant without difficult manipulation. The outcomes of the preliminary experiment allowed for selection of the best doses of inoculum and the best mode of application. As mentioned above, *Lactuca sativa* L. was the preferred cultivar among horticultural crops due to its short cycle. Successful applications have been already reported in different lettuce cultivars with different PGPR strategies [32,35,36,37]. The single inoculum of *Bacillus* sp. VWC18 evidenced a weak response. In particular, it influenced foliar biomass and mineral uptake to some extent (with a reasonable increase), though not significantly for all tested doses. The lowest concentration (10^3^ CFU·mL^−1^) was the one that worked best, but a single dose was not enough to maintain the PGPR effect for the whole crop cycle. Microbial survival and colonization of the rhizosphere depends on several mechanisms that involve both plants and bacteria. Microorganisms in the soil rhizosphere create an interaction with the root surface, stimulating many physical and biochemical activities that can increase plant/root system development, as has been pointed out in other studies [14,38]. Stopping the experiment at that point would have led to wrong conclusions. On the contrary, the parallel experiment with the multiple inoculum allowed for better performing results to be obtained; leaf fresh weight and dry weight for all treatments had a significant increase of 42% and 22%, respectively. The increase in growth parameters was consistent with other authors [32,36], although different *Bacillus* strains were used. Treated plants had an almost twofold increase in foliar mass compared to control plants, thus providing higher market value due to more vigorous plants. All applied concentrations were successful, but the lowest and the highest ones provided the most significant effects. This confirmed that rhizosphere enrichment should be deeply studied by looking at different strategies to optimize the best solution. Microorganisms in the rhizosphere can show infinite kinds of actions; *Bacillus* sp. VWC18 may influence root and leaf development, in particular towards phyto-hormone production or improved mineral availability resulting from root development, as has been reported in previous studies with different *Bacillus* strains [32,33,34]. Mineral uptake is fundamental in increasing the nutraceutical value of edible plants because of the role of macro- and micronutrients in regulating biochemical processes within human cells [39]; therefore, accumulation into green leaves is very important when aiming to enhance food quality and contribute to the human diet. The multiple microbial inoculum strongly improved mineral adsorption at the leaf level; the increase was notably high compared to control plants and the single inoculum. This suggests that the microbial strains carried out the PGP functions as a fertilizer/biostimulant when added to soil at scheduled times [29]. Increases in K, P, and Zn were also observed by Kasozi et al. [33] in lettuce grown in a small-scale aquaponics system following the addition twice a week of a commercial *Bacillus*-based product. The second experiment, with the additional analysis, confirmed and allowed for better characterization of *Bacillus* sp. VWC18 activity within the rhizosphere. The two selected doses of *Bacillus* sp. VWC18 (10^3^–10^9^ CFU·mL^−1^) were applied consistently (every ten days) until the end of the growth cycle. All parameters (leaf and root FW and DW and chlorophyll and mineral concentrations) significantly improved in the treated plants compared to the control, both in lettuce and basil. With respect to lettuce, the increases in biomass and minerals were comparable to those reported previously, although the lettuce seedlings were slightly bigger in the second experiment carried out in 2022. What was surprising was the huge increase in root FW and root DW in both treatments and both crops, confirming the crucial role of the microbial inoculum in root expansion [40,41,42]. Inoculation with *Bacillus* spp. was often reported to lead to higher root development [43,44], and 3′-indole-acetic acid (auxin group) seemed to be the most common phytohormone acting as a promoter of lateral roots and root hair cells [45,46]. Arkhipova et al. [7] observed an accumulation of cytokinins in lettuce plants inoculated with 5 mL of *Bacillus subtilis* strains and an increase in plant shoot and root weight of approximately 30% over 8 days. *Bacillus* sp. VWC18 directly intensifies root development and consequently increases water and mineral adsorption through auxin-like compound production and release. The trend in root DM% in lettuce and basil plants clearly showed a higher water accumulation in roots receiving the microbial inoculum. Moreover, root FW progressively increased with increased amounts of the microbial inoculum, simulating a dose-dependent effect. Similarly, the chlorophyll and carotenoids concentrations showed a dose-dependent increase. Higher chlorophyll production was also reported by Heidari et al. [47] for *Bacillus lentus* in basil plants. Poor root colonization can be a reason for the inconsistent activity of bacterial inoculants in the rhizosphere [48]. Even though not directly investigated, the obtained data show that the persistence/colonization of the microbial inoculum led to greater root development and, as consequence, to higher nutrient availability for plants, which means more nitrogen and magnesium (which are among the essential nutrients for photosynthetic pigments) [49,50]. Moreover, greater root expansion is important when seeking to improve a plant’s capability to respond to water shortages and optimize the use of available water. Although the crop species is different, Aolei et al. [51] reported a 29%, 20%, and 232% increase in root length, root surface area, and lateral root number in perennial ryegrass treated with a *Bacillus* strain. Moreover, total nitrogen and phosphorous increased by 7% and 114%, respectively. In basil, the dose-dependent effect was more pronounced; FW and DW in both treatments had a significant increase that exceeded 200% in both leaves and roots. As observed for lettuce, the microbial inoculum in basil plants would have promoted the production of biologically active substances such as phytohormones, amino acids, and water-soluble vitamins, thus enabling plant cell growth/division and the extension of roots [52]. The nutrient and mineral content in basil plants increased progressively, and the higher *Bacillus* dose caused a strong booster effect. It is undoubtedly a synergistic effect. The obtained results are extremely important considering that basil is often consumed fresh, without cooking. Bacillus can increase the levels of these plant nutrients. Application of microbial, fungal bioactive compounds and biopolymer components in various bioformulations was found to differentially modify the agronomic characteristics and the metabolic profile of basil plants while in some cases increasing the quantity of phenolic compounds, thus producing a qualitatively superior final product. In fact, phenolics strongly contribute to basil antioxidant capacity and biological properties [45,46,47]. Thus, through an increase in these components, well known for their many applications in the field of human health [11,12,13,15,16], the consumer can obtain a more valuable basil plant with enhanced health properties. The second set of experiments clearly showed how the right microbial dose and frequency might increase nutrient and water uptake by acting specifically in different crops. The treated plants were bigger in size with darker green leaves.

## 4. Materials and Methods

### 4.1. Preliminary Assay on Lactuca sativa *L.*

The preliminary experiment was carried out in a protected greenhouse located in Vivaio Ortofloricolo Paganotto (Verona, Italy). Homogenous lettuce seedlings (*Lactuca sativa* L.), belonging to the green typology “Gentilina” with two leaves, were provided by Orto Mio s.r.l. (Verona, Italy) 20 days after sowing and transplanted to individual pots (2L capacity). Pots were filled with the same substrate composed of acid peat, humified peat, and non-composted green soil conditioner (pH 7.5; total porosity *v*/*v*: 80%; dry bulk density of 450 kg·m^2 −1^; provided by Gramoflor gmbh & Co., Vechta, Germany). The experiment consisted of a randomized complete block design that included the microbial treatment at four different doses: B1 (10^3^ CFU·mL^−1^), B2 (10^5^ CFU·mL^−1^), B3 (10^7^ CFU·mL^−1^), and B4 (10^9^ CFU·mL^−1^). The application frequency was established as (S) single inoculum (one-step inoculation of *Bacillus* sp. VWC18 after transplantation) and (M) multiple inoculum (inoculation of *Bacillus* sp. VWC18 after transplantation) every 10 days until harvest. There were 10 pots for each inoculation treatment (S-B1, S-B2, S-B3, and S-B4 and M-B1, M-B2, M-B3, and M-B4) and 20 pots as control (S-CTR and M-CTR, 10 pots each), totaling 100 pots. A crop cycle length of 33 days from transplantation to harvest was adopted.

### 4.2. Greenhouse Experimental Design for Lettuce and Basil

Twenty-day-old lettuce (*Lactuca sativa* L.) plantlets, as indicated above, and forty-day-old basil seedlings (Orto Mio s.r.l., Verona, Italy) belonging to the typology “Genovese” (*Ocimum basilicum* L. cv. Superbo) were grown in the same place and conditions mentioned in Section 4.1. The experimental design consisted of a randomized complete block design with three treatments for each crop and three replicates (6 pots/each replicate). Based on previous results, the technical design included the two best performing microbial doses and the multiple inoculum (at transplanting and every 10 days until harvest) as follows: a) L-CTR: lettuce control plants; b) L-B1: lettuce plants treated with *Bacillus* sp. VWC18 (10^3^ CFU·mL^−1^ ); c) L-B2: lettuce plants treated with *Bacillus* sp. VWC18 (10^9^ CFU mL^−1^ each plant); d) B-CTR: basil control plants; e) B-B1: basil plants treated with *Bacillus* sp. VWC18 (10^3^ CFU·mL^−1^); f) B-B2: basil plants treated with *Bacillus* sp. VWC18 (10^9^ CFU mL^−1^). A total of 54 plants for lettuce (L-CTR, L-B1, L-B2) and 54 for basil (B-CTR, B-B1, B-B2) were set up.

### 4.3. Bacterial Strain Isolation and Molecular Identification

*Bacillus* sp. VWC18 was isolated from vegetable compost waste in piles in an open field (April 2018, Bologna, Emilia Romagna). The temperature of the compost was 75 °C at ∼30 cm depth, and sampling material was taken from between the surface and a depth of ∼20 cm. Compost samples were serially diluted and plated on Tryptic Soy agar (TSA, Oxoid, ThermoFisher, Waltham, MA, USA). Plates were incubated for 2–3 days at 30 °C in aerobic conditions. Isolated bacilli strains were then identified through 16S rRNA gene sequencing [53]. Genomic DNA extraction was performed using the Wizard^®^ Genomic DNA Purification Kit (Promega, Madison, WI, USA). Amplification of the 16S rRNA gene was performed using the universal bacterial primer 8F (5′-AGA GTT TGA TCC TGG CTC AG-3′) and 1520R (5′-AAG GGA GGT GAT CCA GCC GCA-3′) under the following conditions: denaturation at 95 °C for 2 min followed by 30 cycles of 95 °C for 15 s, 57 °C for 1 min, and 72 °C for 1 min and a final extension at 72 °C for 10 min. The amplified PCR product was analyzed using agarose gel electrophoresis (Invitrogen, Waltham, MA, USA) and visualized with the gel documentation system Gel DocTM XR (Bio-Rad, Hercules, CA, USA). The PCR product was purified using NucleoSpin (Macherey-Nagel GmbH & Co. KG, Düren, Germany). The purified PCR product was delivered to Eurofins MWG Operon (Ebersberg, Germany) for sequencing. Received chromatograms were edited and analyzed using the software program Finch TV version 1.4.0 (Geospiza Inc., Seattle, WA, USA). DNAMAN software (Version 6.0, Lynnon BioSoft. Inc., San Ramon, CA, USA) was used to obtain consensus sequences that were processed for the genera/species and accession number assignment was provided on 15 December 2022 (AN: through the GenBank database (NCBI) using the nucleotide BLAST (Basic Local Alignment Search Tool; http://www.ncbi.nlm.nih.gov/BLAST/ (accessed on 13 February 2023)).

### 4.4. Inoculum Setup

*Bacillus* sp. VWC18 was grown in 50 mL of Tryptone Soy Broth (TSB, Oxoid, ThermoFisher) for 20 h at 28 °C under horizontal shaking. Afterward, the microbial biomass was inoculated in 1L of TSB and grown at 28 °C until an OD of 1.0, corresponding to 2 × 10^9^ colony forming unit (CFU)·mL^−1^. The correspondence between OD and CFU was defined following plate counts that were performed three times (data not shown). Bacteria were collected via centrifugation for 10 min at 5000 rpm and then diluted in 100 mL of sterile deionized water to reach the different selected inoculation rates. Treatments were performed through watering, with 10 mL added to the base of each plant to reach a final concentration of 10^4^, 10^6^, 10^8^, and 10^10^ CFU in each pot. The same volume of tap water was added to the control plants, without any other nutrients or the microbial inoculum being applied.

### 4.5. Sampling and Yield Assessment

#### 4.5.1. First Experiment Design: Lettuce

In February 2021, 33 days after transplanting (10 days from the last inoculum), six plants from each treatment and control (60 plants in total) were randomly selected and transported to the laboratory. Plants were cut at the base within 4 h; fresh weight (FW) was determined by weighing the lettuce head immediately after cutting. The aboveground material was then placed in a paper bag and dried at 70 °C for 72 h. The difference in weight before and after drying was used to calculate the shoot dry weight of the sample (DW). The percentage of dry matter (DM%) was calculated as DW/FW × 100. Next, the dried samples were ground into powder in a Willey-type mill and stored until further analysis.

#### 4.5.2. Second Experiment Design: Lettuce and Basil

In March 2022, 31 (lettuce), 54 (basil) days after transplanting, and 10 days after the last treatment, five plants for each replicate and treatment (15 plants for each crop) were randomly selected and transported to the laboratory. Determination of FW, DW, and DM% was performed as mentioned in Section 4.5.1. Measurements of root fresh weight (FW) and dry weight (DW) were also performed, as well as the corresponding dry matter percentage (DM%) calculations.

### 4.6. Chlorophylsl and Carotenoids Determination

In order to determine chlorophyll-a, chlorophyll-b, and carotenoids concentration, the leaf samples of three plants not subjected to drying were ground in acetone and centrifuged, with absorbance at 665 nm, 649 nm, and 470 nm then measured in the supernatant according to the methods of Porra et al. [54] and Lichtenthaler et al. [55].

### 4.7. Mineral Analyses

Dried samples of lettuce and basil (six plant-treatment in the preliminary experiment; five plant-replicate crop in the second experiment) were analyzed for macro- and micronutrients after wet digestion according to US EPA Method 3052 [56].A total of 0.250–0.300 mg DW of each sample was treated with 8 mL of nitric acid (65%) and 2 mL of hydrogen peroxide (30%) at 180 °C in an Ethos TC microwave lab station (Milestone, Bergamo, Italy), and mineral concentration was assessed via inductively coupled plasma optical emission spectroscopy (ICP-OES) (Ametek Spectro Arcos EOP, Kleve, Germany). The following nutrients were investigated: phosphorous (P), potassium (K), sodium (Na), calcium (Ca), magnesium (Mg), manganese (Mn), iron (Fe), zinc (Zn), copper (Cu), boron (B), chrome (Cr), molybdenum (Mo), and selenium (Se). N was determined using the Kjeldahl method [56] by mineralizing 0.300 g of sample with 10 mL of sulfuric acid (95%) and a catalyst (a mix of nitrogen-free anhydrous potassium sulphate and nitrogen-free anhydrous copper sulphate) at 420 °C for 180 min, and subsequent distillation was performed with a 32% NaOH solution (32%) before titration with 0.1 M sulfuric acid.

### 4.8. Statistical Analyses

Experimental data from leaves and roots (FW, DW and DM%), as well as chlorophylls, carotenoids, and macro-microelement concentrations, were analyzed separately using one-way ANOVA and HSD Tukey’s post-hoc analysis at *p* ≤ 0.05, which was performed using R software (www.r-project.org).

## 5. Conclusions

The present study demonstrated that *Bacillus*. sp. VWC18 promoted growth in lettuce and basil plants when constantly applied at targeted doses. Moreover, mineral uptake and chlorophyll and carotenoid concentrations greatly increased, providing higher nutritional value for human consumption. The study also showed that not all applied doses exerted the same effects, which was also true in terms of the frequency of inoculum application. The two selected doses had a different impact in lettuce and basil, thus showing how microbial–plant interaction could be specific. Overall, the lower dose may be more convenient and cost-effective than the higher dose. However, application in fields may not be as relevant as it appears to be for greenhouse and plant factories, where application of the suggested doses represents a green solution that can improve quality and yield commercial plants. The higher dose is extremely effective in terms of root development and probably the best solution for field crops. This research field is very promising, though optimization of the efficacy of this microorganism should be confirmed by performing studies on biochemical features and formulations, as well as investigations into other crop species.

## Figures and Tables

**Figure 1 plants-12-01700-f001:**
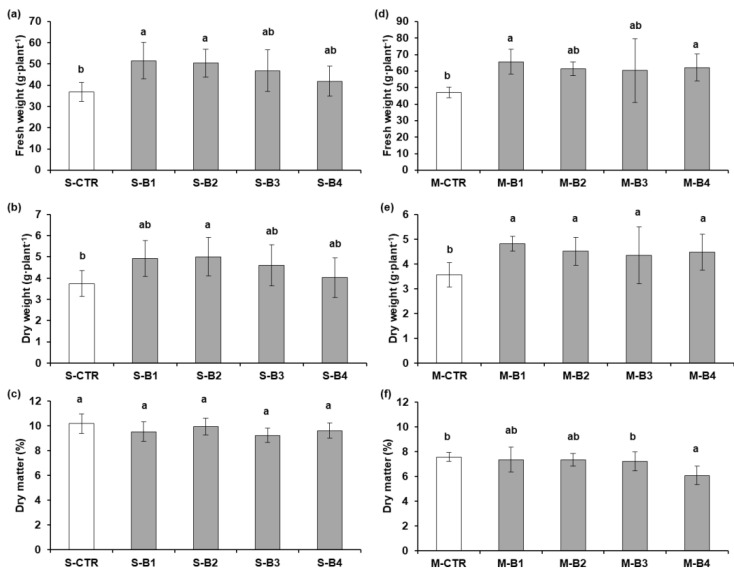
Preliminary greenhouse experiment: growth parameters of lettuce plants untreated and treated with *Bacillus* sp. VWC18 with single (S) and multiple (M) inocula at four different doses (B1 (10^3^ CFU·mL^−1^), B2 (10^5^ CFU·mL^−1^), B3 (10^7^ CFU·mL^−1^), and B4 (10^9^ CFU·mL^−1^)). S-CTR and M-CTR: control plants with single and multiple inocula. Histograms represent leaf biomass: fresh weight (**a**), dry weight (**b**), and dry matter% (**c**) for the single inoculum (S) and (**d**) fresh weight, (**e**) dry weight, and (**f**) dry matter% for the multiple inoculum (M). Different letters (^a,b^) within each graph show significant differences (HSD Tukey’s test) at *p* ≤ 0.05.

**Figure 2 plants-12-01700-f002:**
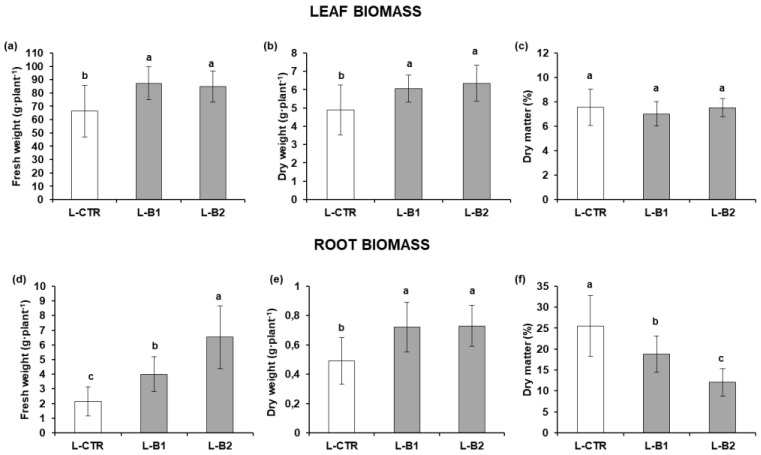
Growth parameters of lettuce plants untreated and treated with *Bacillus* sp. VWC18 at two selected doses: L-B1 (10^3^ CFU·mL^−1^) and L-B2 (10^9^ CFU·mL^−1^). L-CTR: control plants. Histograms represent leaf biomass (**a**) fresh weight, (**b**) dry weight, and (**c**) dry matter% and root biomass (**d**) fresh weight, (**e**) dry weight, and (**f**) dry matter%. Different letters (^a,b,c^) indicate significant differences (HSD Tukey’s test) at *p* ≤ 0.01.

**Figure 3 plants-12-01700-f003:**
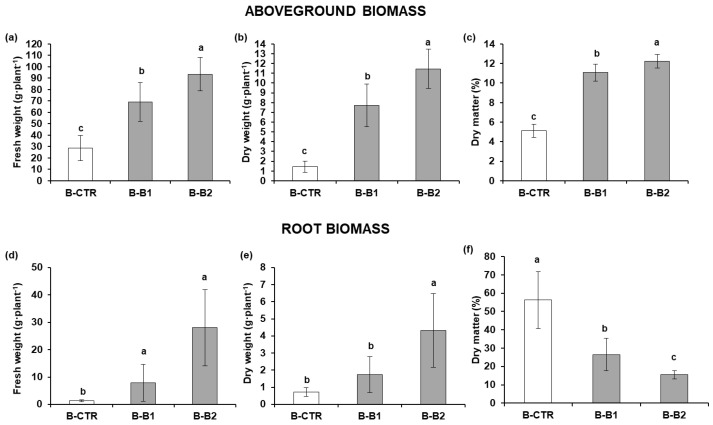
Growth parameters of basil plants untreated and treated with *Bacillus* sp. VWC18 at two selected doses: B-B1 (10^3^ CFU·mL^−1^) and B-B2 (10^9^ CFU·mL^−1^). B-CTR: control plants. Histograms represent leaf biomass (**a**) fresh weight, (**b**) dry weight, and (**c**) dry matter% and root biomass (**d**) fresh weight, (**e**) dry weight, and (**f**) dry matter%. Different letters (^a,b,c^) within each graph indicate significant differences (HSD Tukey’s test) at *p* ≤ 0.01.

**Table 1 plants-12-01700-t001:** Nitrogen (N), phosphorus (P), and potassium (K) content (mg·plant^−1^) in lettuce plants untreated and treated with single (S) and multiple (M) inocula of *Bacillus* sp. VWC18.

Single Inoculum				Multiple Inoculum			
	N	P	K		N	P	K
(mg·Plant^−1^)				(mg·Plant^−1^)			
S-CTR	90.92 ± 6.47 ^a^	8.96 ± 3.22 ^a^	57.18 ± 19.28 ^a^	M-CTR	116.97 ± 38.79 ^b^	12.99 ± 3.62 ^b^	57.15 ± 13.87 ^b^
S-B1	99.37 ± 21.79 ^a^	14.67 ± 5.12 ^a^	91.30 ± 26.72 ^a^	M-B1	165.66 ± 19.87 ^a^	34.49 ± 4.66 ^a^	136.41 ± 11.50 ^a^
S-B2	97.63 ± 11.30 ^a^	14.98 ± 3.74 ^a^	95.27 ± 27.69 ^a^	M-B2	164.52 ± 11.60 ^a^	30.32 ± 5.86 ^a^	117.42 ± 0.51 ^a^
S-B3	99.75 ± 19.14 ^a^	13.21 ± 3.81 ^a^	86.61 ± 30.66 ^a^	M-B3	158.88 ± 26.13 ^a^	26.97 ± 5.97 ^a^	118.46 ± 39.89 ^a^
S-B4	97.54 ± 11.89 ^a^	12.71 ± 2.29 ^a^	78.33 ± 14.27 ^a^	M-B4	170.49 ± 18.39 ^a^	26.81 ± 2.17 ^a^	129.05 ± 19.26 ^a^
*s*	ns	ns	ns		**	***	***

Mean values (±SD): B1, 10^3^ CFU·mL^−1^; B2, 10^5^ CFU·mL^−1^; B3, 10^7^ CFU·mL^−1^; B4, 10^9^ CFU·mL^−1^. S-CTR and M-CTR: control plants for the single and multiple inoculum, respectively. Different letters within each column indicate significant differences according to Tukey’s post-hoc test (*p* ≤ 0.05). ns, **, ***: non-significant or significant at *p* ≤ 0.01 and 0.001, respectively.

**Table 2 plants-12-01700-t002:** Mineral content of lettuce plants untreated and treated with the single (S) and multiple (M) inoculum of *Bacillus* sp. VWC18.

**Single Inoculum (mg·plant^−1^)**								
	**Na**	**Ca**	**Mg**	**Mn**	**Fe**	**Zn**	**Cu**	**B**
S-CTR	10.56 ± 3.48 ^b^	14.63 ± 5.29 ^a^	6.06 ± 2.15 ^a^	0.78 ± 0.25 ^a^	0.29 ± 0.06 ^a^	0.14 ± 0.06 ^a^	0.017 ± 0.005 ^a^	0.18 ± 0.05 ^a^
S-B1	16.05 ± 4.86 ^a^	24.67 ± 7.51 ^a^	9.76 ± 3.10 ^a^	1.31 ± 0.39 ^a^	0.36 ± 0.12 ^a^	0.20 ± 0.08 ^a^	0.022 ± 0.04 ^a^	0.26 ± 0.07 ^a^
S-B2	14.59 ± 1.92 ^ab^	25.88 ± 8.46 ^a^	9.74 ± 2.09 ^a^	1.40 ± 0.62 ^a^	0.37 ± 0.12 ^a^	0.20 ± 0.06 ^a^	0.025 ± 0.07 ^a^	0.25 ± 0.06 ^a^
S-B3	12.75 ± 2.09 ^ab^	22.52 ± 7.06 ^a^	8.50 ± 1.98 ^a^	1.26 ± 0.42 ^a^	0.43 ± 0.10 ^a^	0.22 ± 0.07 ^a^	0.026 ± 0.09 ^a^	0.21 ± 0.05 ^a^
S-B4	13.60 ± 2.32 ^ab^	21.51 ± 4.94 ^a^	8.63 ± 1.64 ^a^	1.18 ± 0.35 ^a^	0.35 ± 0.07 ^a^	0.20 ± 0.04 ^a^	0.022 ± 0.03 ^a^	0.21 ± 0.04 ^a^
*s*	*	ns	ns	ns	ns	ns	ns	ns
**Multiple Inoculum (mg·plant^−1^)**								
	**Na**	**Ca**	**Mg**	**Mn**	**Fe**	**Zn**	**Cu**	**B**
M-CTR	9.01 ± 2.11 ^c^	18.47 ± 3.23 ^b^	7.26 ± 1.60 ^c^	0.74 ± 0.14 ^c^	0.30 ± 0.06 ^b^	0.20 ± 0.06 ^b^	0.024 ± 0.046 ^b^	0.14 ± 0.03 ^c^
M-B1	17.82 ± 1.60 ^a^	33.93 ± 4.41 ^a^	14.63 ± 1.73 ^ab^	1.72 ± 0.27 ^ab^	0.60 ± 0.08 ^a^	0.41 ± 0.06 ^a^	0.046 ± 0.006 ^a^	0.29 ± 0.04 ^a^
M-B2	14.40 ± 2.35 ^b^	28.94 ± 5.08 ^a^	12.75 ± 2.38 ^ab^	1.60 ± 0.32 ^ab^	0.54 ± 0.13 ^a^	0.36 ± 0.07 ^a^	0.040 ± 0.008 ^a^	0.25 ± 0.05 ^ab^
M-B3	13.30 ± 2.13 ^b^	27.70 ± 11.76 ^a^	12.05 ± 3.39 ^b^	1.27 ± 0.41 ^bc^	0.50 ± 0.16 ^ab^	0.33 ± 0.07 ^a^	0.040 ± 0.008 ^a^	0.22 ± 0.04 ^b^
M-B4	15.60 ± 1.76 ^ab^	37.48 ± 3.90 ^a^	15.98 ± 1.58 ^a^	1.81 ± 0.47 ^a^	0.62 ± 0.10 ^a^	0.43 ± 0.04 ^a^	0.046 ± 0.005 ^a^	0.30 ± 0.03 ^a^
*s*	***	***	***	***	***	***	***	***

Mean values (±SD): B1, 10^3^ CFU·mL^−1^; B2, 10^5^ CFU·mL^−1^; B3, 10^7^ CFU·mL^−1^; B4, 10^9^ CFU·mL^−1^. S-CTR and M-CTR: control plants for the single and multiple inoculum, respectively. Different letters within each column indicate significant differences according to Tukey’s post-hoc test (*p* ≤ 0.05). ns, *, ***: non-significant or significant at *p* ≤ 0.05 and 0.001, respectively.

**Table 3 plants-12-01700-t003:** Chlorophyll-*a*, chlorophyll-*b*, and carotenoids concentrations for lettuce plants treated or untreated with *Bacillus* sp. VWC18.

	Chlorophyll-*a* (µg·g^−1^)	Chlorophyll-*b* (µg·g^−1^)	Carotenoids (µg·g^−1^)
L-CTR	42.29 ± 4.42 ^b^	18.54 ± 9.20 ^b^	4310.03 ± 101.84 ^b^
L-B1	103.18 ± 13.68 ^b^	46.15 ± 5.50 ^b^	7147.74 ± 533.33 ^b^
L-B2	335.76 ± 135.55 ^a^	172.23 ± 37.93 ^a^	21174.75 ± 9445.08 ^a^
*s*	**	***	*

Mean values (±SD) for L-CTR (control plants): L-B1, 10^3^ CFU·mL^−1^; L-B2, 10^9^ CFU·mL^−1^. Different letters within each column indicate significant differences according to Tukey’s post-hoc test (*p* ≤ 0.05). *, **, ***: significant at *p* ≤ 0.05, *p* ≤ 0.01, and *p* ≤ 0.001, respectively.

**Table 4 plants-12-01700-t004:** Mineral content of lettuce plants untreated and treated with *Bacillus* sp. VWC18.

	**N**	**P**	**K**	**Na**	**Ca**	**Mg**
**(mg·plant^−1^)**						
L-CTR	168.45 ± 11.45 ^b^	18.18 ± 5.19 ^b^	78.88 ± 36.29 ^b^	11.12 ± 7.56 ^b^	21.46 ± 5.92 ^b^	8.24 ± 2.56 ^c^
L-B1	235.40 ± 30.02 ^a^	35.35 ± 2.65 ^a^	181.09 ± 23.06 ^a^	19.14 ± 7.28 ^a^	34.33 ± 6.23 ^a^	13.37 ± 1.23 ^b^
L-B2	242.38 ± 46.26 ^a^	32.40 ± 3.18 ^a^	183.38 ± 21.27 ^a^	20.12 ± 7.18 ^a^	37.24 ± 3.28 ^a^	15.38 ± 1.56 ^a^
*s*	***	***	**	***	***	***
	**Mn**	**Fe**	**Zn**	**Cu**	**B**	**Cr**
L-CTR	0.80 ± 0.33 ^c^	0.40 ± 0.10 ^b^	0.24 ± 0.08 ^b^	0.028 ± 0.008 ^b^	0.16 ± 0.04 ^b^	0.0040 ± 0.0012 ^b^
L-B1	1.66 ± 0.25 ^b^	0.75 ± 0.15 ^a^	0.44 ± 0.05 ^a^	0.051 ± 0.009 ^a^	0.28 ± 0.03 ^a^	0.0058 ± 0.0011 ^a^
L-B2	1.86 ± 0.09 ^a^	0.74 ± 0.14 ^a^	0.42 ± 0.06 ^a^	0.049 ± 0.009 ^a^	0.32 ± 0.03 ^a^	0.0060 ± 0.0010 ^a^
*s*	***	***	**	***	***	***
	**Mo**	**Se**				
L-CTR	0.0011 ± 0.0009 ^b^	0.0036 ± 0.0008 ^b^				
L-B_1_	0.0019 ± 0.0007 ^ab^	0.0055 ± 0.00021 ^a^				
L-B_2_	0.0021 ± 0.0015 ^a^	0.0051 ± 0.0015 ^a^				
*s*	*	**				

Mean values (±SD): L-B1, 10^3^ CFU·mL^−1^; L-B2, 10^9^ CFU·mL^−1^. Different letters within each column indicate significant differences according to Tukey’s post-hoc test (*p* ≤ 0.05). *, **, ***: significant at *p* ≤ 0.05, 0.01, and 0.001, respectively.

**Table 5 plants-12-01700-t005:** Chlorophyll-*a*, chlorophyll-*b*, and carotenoids concentrations of basil plants treated or untreated with *Bacillus* sp. VWC18.

	Chlorophyll-*a* (µg·g^−1^)	Chlorophyll-*b* (µg·g^−1^)	Carotenoids (µg·g^−1^)
B-CTR	42.26 ± 1.91 ^b^	30.33 ± 0.62 ^b^	3.127 ± 145 ^c^
B-B_1_	120.08 ± 2.52 ^a^	100.62 ± 33.15 ^a^	13.183 ± 896 ^b^
B-B_2_	122.89 ± 4.04 ^a^	127.38 ± 7.20 ^a^	15.292 ± 628 ^a^
*s*	***	**	***

Mean values (±SD): B-B1, 10^3^ CFU·mL^−1^; B-B2, 10^9^ CFU·mL^−1^. B-CTR: control plants. Different letters within each column indicate significant differences according to Tukey’s post-hoc test (*p* ≤ 0.05). **, ***: significant at *p* ≤ 0.01 and *p* ≤ 0.001, respectively.

**Table 6 plants-12-01700-t006:** Mineral content of basil plants untreated and treated with *Bacillus* sp. VWC18.

	**N**	**P**	**K**	**Na**	**Ca**	**Mg**
**(mg·plant^−1^)**						
B-CTR	39.47 ± 5.18 ^c^	9.92 ± 0.99 ^b^	62.40 ± 12.22 ^c^	0.15 ± 0.25 ^c^	15.34 ± 3.99 ^c^	3.90 ± 1.41 ^c^
B-B1	231.15 ± 43.83 ^b^	19.80 ± 4.86 ^a^	304.99 ± 65.36 ^b^	1.32 ± 0.82 ^b^	135.72 ± 40.37 ^b^	26.69 ± 9.10 ^b^
B-B2	372.84 ± 34.84 ^a^	21.02 ± 3.22 ^a^	374.41 ± 76.94 ^a^	7.06 ± 2.05 ^a^	196.72 ± 39.64 ^a^	38.63 ± 8.84 ^a^
*s*	***	***	**	***	***	***
	**Mn**	**Fe**	**Zn**	**Cu**	**B**	**Cr**
**(mg·plant^−1^)**						
B-CTR	0.05 ± 0.2 ^c^	0.16 ± 0.20 ^c^	0.08 ± 0.02 ^c^	0.003 ± 0.001 ^b^	0.03 ± 0.01 ^c^	0.0003 ± 0.00006 ^b^
B-B1	2.55 ± 0.72 ^b^	1.50 ± 0.54 ^b^	0.46 ± 0.15 ^b^	0.095 ± 0.025 ^a^	0.17 ± 0.04 ^b^	0.0186 ± 0.0147 ^ab^
B-B2	3.62 ± 1.05 ^a^	2.47 ± 1.38 ^a^	0.75 ± 0.30 ^a^	0.118 ± 0.039 ^a^	0.24 ± 0.05 ^a^	0.0366 ± 0.0422 ^a^
*s*	***	***	***	***	***	**
	**Mo**	**Se**				
**(mg·plant^−1^)**						
B-CTR	0.016 ± 0.007 ^b^	0.0030 ± 0.002 ^c^				
B-B1	0.016 ± 0.007 ^b^	0.0071 ± 0.003 ^b^				
B-B2	0.054 ± 0.017 ^a^	0.0160 ± 0.003 ^a^				
*s*	***	***				

Mean values (±SD): B-B1, 10^3^ CFU·mL^−1^; B-B2, 10^9^ CFU·mL^−1^. B-CTR: control plants. Different letters within each column indicate significant differences according to Tukey’s post-hoc test (*p* ≤ 0.05). **, ***: significant at *p* ≤ 0.01 and 0.001, respectively.

## Data Availability

Data are contained within the article.

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
