# Peer review of "Yield and Nutraceutical Value of Lettuce and Basil Improved by a Microbial Inoculum in Greenhouse Experiments"

_plants, 2023, doi:10.3390/plants12081700_

Round 1

Reviewer 1 Report

Dear Authors

There are many studies regarding the bacterial effect on vegetable and lettuce (Like Kröber et al. 2014, Arkhipova et al. 2005,....) , what is the novelty of current research? Please highlight novelty and significant of research in the introduction. 

The abstract could be more specific. same in conclusion. 

What is the substrate used in this experiment? 

How many plants per replication?

Best Regards

Reviewer 2 Report

The MS presents some interesting insights for improving nutrient use efficiency but there are many gaps which need to be addressed prior to considering the same:

1. The technical design of the MS is not clear. Please summarize the treatments.

2. Details of identification of microbial isolate is not included in materials and method section. Only few sentences have been included in discussion but it is irrelevant. Please include details of geographical location of isolation, biochemical characterization, details of molecular identification, which species???, etc.

3. Please include precise details of all the methods employed. Only a short overview has been presented.

4. What kind of soil was used for experiments?? 

5. The authors state "Treatments were performed by watering 10 mL to the base of each plant to reach a final concentration of 104, 106, 108, 1010 CFU · mL-1 each pot", this is not a sound methodology for delivery of microorganisms in soil. Please justify why more accepted methods not employed. Secondly, what was the age of the plants when this treatment was performed?? How much time was allowed prior to plant testing for establishing plant microbe interaction??

6. Please italicize the scientific names of organisms through the MS.

7. Please critically revise the MS for omitting typographical and syntax errors.

Round 2

Reviewer 2 Report

Significant changes have been made in the revised manuscript but the following issues need to be reconsidered:

1. The authors need to discuss the results presented in tables 1-6 in the discussion section.

2. Please address the unusually high SD values.

3. Also, explain discuss the uptake mechanisms intercepted by PGPR to to increase the nutrient use efficiency. Lastly, please elaborate on the competitive uptake of mono- and divalent ions by root system. 

Author Response

Reviewer#2

Dear Reviewer,

We improved the MS and we answered point-by-point to your requests.

Significant changes have been made in the revised manuscript but the following issues need to be reconsidered:

The authors need to discuss the results presented in tables 1-6 in the discussion section.

We agree with the reviewer. The discussion have been improved by adding some new comments to the discussion section (a reference N° 51 has been added); most of the articles dealing with Bacillus and application on lettuce/basil have been already cited in the MS (see references from N° 33). When different crops and different strategies (single and multiple inoculum) are used, it is hard to discuss all the results, and information in litterature is scarce. We reported in the text that the results (growth parameters and mineral uptake) are consistent with other authors; however, most of the experiments are not comparable in terms of PGPR strategy or type of cultivar. This is also reported in the discussion.

Please address the unusually high SD values.

We observed in some results a high SD value, in particular in growth parameters, but measurements were all subjected to one-way analysis of variance (ANOVA) and values not significant were all registered.

Also, explain discuss the uptake mechanisms intercepted by PGPR to increase the nutrient use efficiency. Lastly, please elaborate on the competitive uptake of mono- and divalent ions by root system. 

We focused the discussion on the ability of Bacillus WVC18 to improve root expansion and increase the nutrient adsorption (this is what we really observed). We hypothized a auxin-like compounds production and release and we reported different references to suppor our hypothesis (40: Dal Cortivo et al., 2018; 41: De Aquino et al., 2018; 42: Rojas-Padilla et al., 2022; 43: Lim et al., 2016; 44: Medeiros et al., 2021; 7: Arkhipova et al., 2005).

Round 3

Reviewer 2 Report

Authors have duly incorporated the comments and suggestions.